# Simulating Finite-Time Isothermal Processes with Superconducting Quantum Circuits

**DOI:** 10.3390/e23030353

**Published:** 2021-03-16

**Authors:** Jin-Fu Chen, Ying Li, Hui Dong

**Affiliations:** 1Beijing Computational Science Research Center, Beijing 100193, China; chenjinfu@csrc.ac.cn; 2Graduate School of China Academy of Engineering Physics, No. 10 Xibeiwang East Road, Haidian District, Beijing 100193, China; yli@gscaep.ac.cn

**Keywords:** quantum thermodynamics, quantum circuit, open quantum system, isothermal process, IBM quantum computer

## Abstract

Finite-time isothermal processes are ubiquitous in quantum-heat-engine cycles, yet complicated due to the coexistence of the changing Hamiltonian and the interaction with the thermal bath. Such complexity prevents classical thermodynamic measurements of a performed work. In this paper, the isothermal process is decomposed into piecewise adiabatic and isochoric processes to measure the performed work as the internal energy change in adiabatic processes. The piecewise control scheme allows the direct simulation of the whole process on a universal quantum computer, which provides a new experimental platform to study quantum thermodynamics. We implement the simulation on ibmqx2 to show the 1/τ scaling of the extra work in finite-time isothermal processes.

## 1. Introduction

Quantum thermodynamics, originally considered an extension of classical thermodynamics, has sharpened our understanding of the fundamental aspects of thermodynamics [1,2,3,4,5,6]. Along with the theoretical progress, experimental tests and validations of the principles are relevant in the realm. Simulation of the quantum thermodynamic phenomena [7,8,9,10], as one of the experimental efforts, has been intensively explored with specific systems, e.g., a single trapped particle for testing the Jarzynski equation [11,12], the trapped interacting Fermi gas for quantum work extraction [13,14], and the superconducting qubits for the shortcuts to adiabaticity [15,16]. These specific systems often have limited functions to test generic quantum thermodynamic properties. In quantum thermodynamics, the concerned system, as an open quantum system, generally evolves with the coupling to the environment. Simulations of open quantum systems have been proposed theoretically in terms of quantum channels [17,18,19,20,21], and realized experimentally on various systems, e.g., trapped ions [22], photons [23], nuclear spins [24], superconducting qubits [25], and IBM quantum computer recently [26,27]. The previous works mainly focus on simulating fixed open quantum systems, where the parameters of the systems are fixed with the evolution governed by a time-independent master equation. To devise a quantum heat engine, it is necessary to realize tuned open quantum systems to formulate finite-time isothermal processes.

Simulation with generic quantum computing systems shall offer a universal system to demonstrate essential quantum thermodynamic phenomena. Yet, simulation of a tuned open quantum system remains a challenge mainly due to the inability to physically tune the control parameters and the difficulty to measure the work extraction. In quantum thermodynamics, the work extraction, as a fundamental quantity [28,29,30], requires the tuning of the control parameters. Such requirement is achievable in the specifically designed system, e.g., the laser-induced force on the trapped ion [11], the trapped frequency of the Fermi gas [13,14], and the external field in the superconducting system [15,16]. However, on a universal quantum computer, e.g., IBM quantum computer (ibmqx2), the user is forbidden to tune the actual physical parameters since the parameters have been optimized to reduce errors. An additional problem is the measurement of the work extraction. In classical thermodynamics, it is obtained by recording the control parameters and measuring the conjugate quantities, but such measurement is not suitable in the quantum region [31].

In this paper, an experimental proposal is given to overcome the difficulty in simulating a finite-time isothermal process. We introduce a virtual way to tune the control parameters, i.e., without physically tuning the parameters. The dynamics are realized by quantum gates encoded the parameter change. As a demonstration, we realize the simulation of a two-level system on ibmqx2 [32] for the isothermal processes, which are fundamental to devise quantum heat engines, yet complicated due to the coexistence of the changing Hamiltonian and the interaction with the thermal bath.

To implement the simulation on a universal quantum computer, we adopt a discrete-step method to approach the quantum isothermal process [33,34,35,36,37,38]: the isothermal process is divided into series of elementary processes, each consisting of an adiabatic process and an isochoric process. In the adiabatic process, the parameter tuning is performed virtually with the unitary evolution implemented by quantum gates. In the isochoric process, the dissipative evolution is carried out with quantum channel simulation [23,25,39,40,41,42] with ancillary qubits, which play the role of the environments [18,21,26]. With this approach, we achieve the simulation of the isothermal process on the generic quantum computing system without physically tuning the control parameters. The piecewise control scheme distinguishes work and heat, which are separately generated and measured as the internal energy change in the two processes. In the current simulation, the energy spacing of the two-level system is tuned with the unchanged ground and excited states. The tuning of the energy spacing is virtually performed via modulating the thermal transition rate in the isochoric process.

In our proposal, the simulation with a universal quantum computer brings clear advantages. First, *the arbitrary change of the control parameters* is archived by the virtual tuning via the simulation of corresponding dynamics, avoiding the difficulty in tuning the actual physical system. In turn, the parameters can be controlled to follow an arbitrary designed function. Second, we can realize *the immediate change of environmental parameters*, such as the temperature. The effect of the bath is reflected through the state of the auxiliary qubits, which can be controlled flexibly with quantum gates.

## 2. Discrete-Step Method to Quantum Isothermal Process

In quantum thermodynamics, the concerned system generally evolves under the changing Hamiltonian while in contact with a thermal bath. The interplay between quantum work and heat challenges to characterize the quantum thermodynamic cycle on the microscopic level, where the classical method to measure the work via force and distance, is not applicable [31]. For the timescale of the tuning far smaller than the thermal bath response time, the evolution is thermodynamic adiabatic, where the heat exchange with the thermal bath can be neglected, and the internal energy changes due to the performed work through the changing control field. The opposite extreme case with the unchanged control parameters is known as the isochoric process, where the internal energy changes are induced by the heat exchange with the thermal bath. Therefore, work and heat are separated clearly in the adiabatic and isochoric processes, and are obtained directly by measuring the internal energy change.

To simulate the general processes on a universal quantum computer, a piecewise control scheme is necessary to express the continuous non-unitary evolution in terms of quantum channels, where the evolution in each period is constructed by the simulations of open quantum systems [21]. To separate work and heat, we adopt the discrete-step method by dividing the whole process into series of piecewise adiabatic and isochoric processes [33,34,35,36,37,38]. In Figure 1, the discrete-step method is illustrated with the minimal quantum model, a two-level system with the energy spacing ω(t) between the ground state $|g\rangle$ and the excited state $|e\rangle$. Such a two-level system can be physically realized with a qubit, as an elementary unit of the quantum computer. For the clarity of the later discussion, we use the term “two-level system” to denote the simulated system and “qubit” as the simulation system hereafter without specific mention.

The state of the two-level system is represented by the density matrix $\rho_{s}\left(t\right)$ of the system qubit, and the thermal bath is simulated by ancillary qubits. Initially, the system qubit is prepared to the thermal state $\rho_{s}\left(0\right)$ at the temperature *T*. The evolution of the tuned open quantum system is implemented with single-qubit and two-qubit quantum gates. The internal energy of the two-level system is $E\left(t\right)=\omega \left(t\right)p_{e}\left(t\right)$, where the energy of the ground state is assumed as zero, and the population in the ground (excited) state is $p_{g}\left(t\right)=\langle g|\rho_{s}\left(t\right)|g\rangle$ ($p_{e}\left(t\right)=\langle e|\rho_{s}\left(t\right)|e\rangle$).

For the system to be simulated, we use the discrete-step method to approach the finite-time isothermal process for the two-level system. The discrete isothermal process contains *N* steps of elementary processes with the total operation time $\tau +\tau_{\mathrm{adi}}$, where τ ($\tau_{\mathrm{adi}}$) denotes the operation time in the isochoric (adiabatic) process. Each elementary process is composed of an adiabatic and an isochoric processes. We set the equal operation time for every elementary process $\delta \tau =(\tau +\tau_{\mathrm{adi}})/N$, with the duration τ/N ($\tau_{\mathrm{adi}}/N$) for each isochoric (adiabatic) process.

In the adiabatic process, the system is isolated from the thermal bath and evolves under the time-dependent Hamiltonian. Such a process is described by a unitary evolution with the time $\tau_{\mathrm{adi}}/N$. The performed work is determined by the change of the internal energy at the initial and the final time. For a generic adiabatic process, the unitary evolution of the system can be simulated with the single-qubit gate acted on the system qubit. In this paper, we consider the adiabatic process as the quench with zero time $\tau_{\mathrm{adi}}=0$, occurred at time $t_{j-1}=\left(j-1\right)\delta \tau ,\;j=1,2,...,N.$ As the result of the quench, the energy spacing is shifted from $\omega_{j-1}$ to $\omega_{j}$, while the density matrix $\rho_{s}\left(t_{j-1}\right)$ remains unchanged after the quench. At the initial time $t_{0}=0$, the energy is quenched from $\omega_{0}$ to $\omega_{1}$ after the initial preparation. The performed work for the quench at time $t_{j-1}$ reads
(1)$$W_{j}=\left(\omega_{j}-\omega_{j-1}\right)p_{e}\left(t_{j-1}\right).$$

To obtain the performed work, we only need to measure the excited state population $p_{e}\left(t_{j-1}\right)$ of the system qubit at the beginning of each isochoric process.

In the isochoric process of the *j*-th elementary process ($t_{j-1}<t\leq t_{j}$), the two-level system is brought into contact with the thermal bath at the temperature *T*. The evolution is given by the master equation
(2)$$\begin{array}{cc} {\dot{\rho }}_{s} & =-i\left[H_{j},\rho_{s}\right]+\gamma_{0}N_{j}L\left(\sigma_{+}\right)\left[\rho_{s}\right]\\ \; & +\gamma_{0}\left(N_{j}+1\right)L\left(\sigma_{-}\right)\left[\rho_{s}\right], \end{array}$$
with
(3)$$L\left(\sigma \right)\left[\rho_{s}\right]=\sigma \rho_{s}\sigma^{†}-\frac{1}{2}\sigma^{†}\sigma \rho_{s}-\frac{1}{2}\rho_{s}\sigma^{†}\sigma .$$

Here, $H_{j}=\omega_{j}|e\rangle \langle e|$ is the Hamiltonian of the system during the period $t_{j-1}<t\leq t_{j}$, $N_{j}=1/\left[\exp\left(\beta \omega_{j}\right)-1\right]$ is the average photon number with the inverse temperature $\beta =1/(k_{B}T)$, and $\sigma_{+}=\;|e\rangle \langle g|$ ($\sigma_{-}=\;|g\rangle \langle e|$) is the transition operator. In this process, the change of the internal energy is induced by the heat exchange with the thermal bath, and no work is performed. During the whole discrete isothermal process, the work is only performed at the time $t_{j}$.

We explicitly give the equations for each element of the density matrix according to Equation (Equation 2). The populations in the ground and excited states satisfy
(4)$${\dot{p}}_{g}=\gamma_{0}\left(N_{j}+1\right)p_{e}-\gamma_{0}N_{j}p_{g},$$
and $p_{e}=1-p_{g}$. The off-diagonal elements $\rho_{eg}\left(t\right)=\langle e|\rho_{s}\left(t\right)|g\rangle$ and $\rho_{ge}\left(t\right)=\langle g|\rho_{s}\left(t\right)|e\rangle$ satisfy
(5)$${\dot{\rho }}_{eg}=-i\omega_{j}\rho_{eg}-\gamma_{0}\left(2N_{j}+1\right)\rho_{eg},$$
and $\rho_{ge}\left(t\right)=\rho_{eg}^{*}\left(t\right)$. With the unchanged energy eigenstates, the diagonal and the off-diagonal elements of the density matrix evolve separately during the whole isothermal process.

## 3. Simulation with Quantum Circuits

In this section, we first show the simulation of one elementary process in the circuit. The simulation is formulated for the adiabatic and the isochoric processes as follows.

**Adiabatic process**. In the superconducting quantum computer, e.g., IBM Q system, the tuning of the physical energy levels of qubits is unavailable for the users. The physical parameters are fixed at the optimal values to possibly reduce noises and errors induced by decoherence and imperfect control.

We consider the Hamiltonian of the simulated two-level system as $H\left(t\right)=\omega \left(t\right)|e\rangle \langle e|$ with the piecewise tuned energy spacing
(6)$$\omega \left(t\right)=\omega_{j},\;t_{j-1}<t\leq t_{j}\;\mathrm{with}\;j=1,2...,N.$$

We will show that the tuning of the energy spacing ω(t) only affects the thermal transition rate. In the simulation, the thermal transition is simulated through the quantum channel simulation, and can be flexibly modulated by single-qubit gates acted on the ancillary qubits. Therefore, we do not have to physically tune any parameters of the quantum computer, and just algorithmically modulate the simulated thermal transition instead. We propose a virtual tuning of the energy spacing with details explained as follows.

In the virtual process, we need to simulate the unitary evolution of the adiabatic process with single-qubit gates acted on the system qubit. For the adiabatic process, i.e., the quench, the state of the system does not evolve in a short period. We just pretend that the energy of the simulated system is tuned from $\omega_{j-1}$ to $\omega_{j}$ in the *j*-th adiabatic process. This virtual tuning of the energy is reflected by the modulation of the transition rate in the simulation of the isochoric process.

**Isochoric process**. The dynamical evolution of the isochoric process can be simulated with the generalized amplitude damping channel (GADC)
(7)$$\rho_{s}\left(t_{j}\right)=E_{\mathrm{GAD}}^{\left(j\right)}\left[e^{-iH_{j}\delta \tau }\rho_{s}\left(t_{j-1}\right)e^{iH_{j}\delta \tau }\right],$$
where $E_{\mathrm{GAD}}^{\left(j\right)}=p_{↓}^{\left(j\right)}E_{↓}^{\left(j\right)}+p_{↑}^{\left(j\right)}E_{↑}^{\left(j\right)}$ is divided into two sub-channels, the amplitude damping channel
(8)$$E_{↓}^{\left(j\right)}\left[\rho_{s}\right]=M_{0}^{\left(j\right)}\rho_{s}M_{0}^{(j)†}+M_{1}^{\left(j\right)}\rho_{s}M_{1}^{(j)†},$$
and the amplitude pumping channel
(9)$$E_{↑}^{\left(j\right)}\left[\rho_{s}\right]=M_{2}^{\left(j\right)}\rho_{s}M_{2}^{(j)†}+M_{3}^{\left(j\right)}\rho_{s}M_{3}^{(j)†}.$$

The corresponding Kraus operators are $M_{0}^{\left(j\right)}=\cos\theta_{j}|e\rangle \langle e|+|g\rangle \langle g|$, $M_{1}^{\left(j\right)}=\sigma_{-}\sin\theta_{j}$, $M_{2}^{\left(j\right)}=|e\rangle \langle e|+\cos\theta_{j}|g\rangle \langle g|$ and $M_{3}^{\left(j\right)}=\sigma_{+}\sin\theta_{j}$. The coefficient $p_{↑}^{\left(j\right)}=1/\left[\exp\left(\beta \omega_{j}\right)+1\right]$ ($p_{↓}^{\left(j\right)}=1-p_{↑}^{\left(j\right)}$) shows the probability of excitation (de-excitation) of the two-level system induced by the thermal bath. The evolution time of the *j*-th elementary process is encoded in the control parameter $\theta_{j}$ via
(10)$$\cos\theta_{j}=\exp\left[-\frac{\gamma_{0}\delta \tau }{2}\coth\left(\frac{\beta \omega_{j}}{2}\right)\right].$$

With infinite operation time, the ideal discrete isothermal process is realized by setting $\theta_{j}=\pi /2$, where the system reaches thermal equilibrium at the end of each isochoric process.

For the initial thermal state $\rho_{s}\left(0\right)=\exp\left(-\beta H\left(0\right)\right)/\mathrm{Tr}\left[\exp\left(-\beta H\left(0\right)\right)\right]$, the off-diagonal element remains zero throughout the whole process in the current control scheme. In this situation, the evolution by Equation (Equation 7) is simplified as
(11)$$\rho_{s}\left(t_{j}\right)=E_{\mathrm{GAD}}^{\left(j\right)}\left[\rho_{s}\left(t_{j-1}\right)\right].$$

For an initial state with non-zero off-diagonal elements, the off-diagonal elements does not affect the evolution of the populations. This comes from the fact that the diagonal and the off-diagonal elements satisfy separate differential equations by Equations (Equation 4) and (Equation 5).

Figure 2 shows the quantum circuit to simulate the isochoric process. The two sub-channels $E_{↓}^{\left(j\right)}$ and $E_{↑}^{\left(j\right)}$ are realized with an ancillary qubit initially prepared in the ground state. The circuits for these two sub-channels are illustrated in Figure 2a. The meaning of each gate is explained at the bottom of Figure 2. Such simulation circuits are extensively studied in the field of quantum computing and quantum information that we will not explain the setup in detail [40].

To achieve the random selection of the two sub-channels, we design two simulation methods, the hybrid simulation, and the fully quantum simulation, as shown in Figure 2b,c, respectively. The former uses one ancillary qubit for each elementary process under the assist of a classical random number generator (CRNG). The latter utilizes fully quantum circuits with two ancillary qubits for each elementary process. In Table 1, we summarize the simulation procedure for the adiabatic and the isochoric processes.

### 3.1. Hybrid Simulation of Isochoric Process with Classical Random Number
Generator (CRNG)

With the limited number of qubits, it is desirable to reduce the unnecessary usage of qubits. For the quantum channel of the system qubit, one ancillary qubit is inevitably needed to simulate the non-unitary evolution of the open quantum system [42]. In this method, one qubit represents the two-level system, and each elementary process adds one more ancillary qubit. Therefore, it requires N+1 qubits to simulate the *N*-step isothermal process.

In the hybrid simulation, the CRNG is used to select the sub-channel $O_{j}^{\left[l\right]}=E_{↑}^{\left(j\right)}$ or $E_{↓}^{\left(j\right)}$ for the isochoric process in the *j*-th elementary process, as shown in Figure 2b. *l* denotes the *l*-th simulation of the discrete isothermal process. For each isochoric process, the CRNG generates a random number $r_{j}^{\left[l\right]}\in \left[0,1\right]$ with uniform distribution. The sub-channel $O_{j}^{\left[l\right]}$ is selected as $E_{↓}^{\left(j\right)}$ ($E_{↑}^{\left(j\right)}$ ) when the random number satisfies $r_{j}^{\left[l\right]}\leq p_{↓}^{\left(j\right)}$ ($r_{j}^{\left[l\right]}>p_{↓}^{\left(j\right)}$).

### 3.2. Fully Quantum Simulation of Isochoric Process

For the system with adequate available qubits, the selection of the two sub-channels can be realized on fully quantum circuit by adding two ancillary qubits for each elementary process, as shown in Figure 2c. In each step, one more ancillary qubit is used, prepared to the super-position state $\cos\left(\alpha_{j}/2\right)|0\rangle +i\sin\left(\alpha_{j}/2\right)|1\rangle$ through the $R_{x}\left(\alpha_{j}\right)$ gate with $\cos\left(\alpha_{j}/2\right)=\sqrt{p_{↓}^{\left(j\right)}}$. This method requires 2N+1 qubits to simulate the *N*-step isothermal process.

Currently, we have solved the problem of separating work and heat. The unitary evolution of the adiabatic process requires isolation from the environment, while the isochoric process needs contact with the environment. Switching on and off the interaction with the thermal bath is complicated and requires enormous efforts, especially in the quantum region for a microscopic system. Fortunately, the design of the quantum computer with a long coherent time ensures the isolation from the environment. The simulation of the quantum channel is designed to simulate the effect of the environment. The advantage of quantum channel simulation over the real coupling to the environment is the flexibility to tune the control parameters, e.g., the temperature, the coupling strength, et al.

The whole evolution of the isothermal process is realized by merging the circuit of each elementary process. In Figure 3, the circuit for the two-step isothermal process is shown as an example. Figure 3a shows the excited state population $p_{e}\left(t\right)$ with the tuned energy spacing ω(t) in a two-step isothermal process. The energy spacing is increased from $\omega_{0}$ to $\omega_{2}$ in two steps, while the excited state population decreases from $p_{0}$ to $p_{2}$.

Figure 3b shows the quantum circuit for the hybrid simulation on ibmqx2. With the five qubits, it is feasible to simulate a four-step isothermal process on ibmqx2. Due to the limited qubit number, the initial state is prepared as a pure state to mimic the thermal state in the current simulation. The populations in the energy eigenstates of the pure state is equal to those of the thermal state, while the non-zero off-diagonal elements lead to the coherence as the superposition of the excited and the ground states. As stated in the description of the isochoric process, such coherence does not affect the evolution of the populations. With another ancillary qubit, a thermal state of the two-level system can be initially prepared through the entanglement between the system and the ancillary qubit.

In the hybrid simulation, the sub-channel $O_{j}^{\left[l\right]}$ of each elementary process is selected as either the amplitude damping $E_{↓}^{\left(j\right)}$ or the pumping one $E_{↑}^{\left(j\right)}$. For an *N*-step isothermal process, there are $2^{N}$ selections of the sub-channels $\left\{O_{1}^{\left[l\right]},O_{2}^{\left[l\right]},...,O_{j}^{\left[l\right]},...O_{N}^{\left[l\right]}\right\}$. The circuit of each selection with N=2,3 and 4 is implemented on ibmqx2. For each selection, the excited state population $p_{e}^{\left[l\right]}\left(t_{j}\right)$ at each step is obtained by repeated implementations of the corresponding circuit. The work in each selection, namely the microscopic work, is given by
(12)$$W^{\left[l\right]}=\sum_{j=1}^{N}\left(\omega_{j}-\omega_{j-1}\right)p_{e}^{\left[l\right]}\left(t_{j-1}\right).$$

The performed work $\bar{W}$ of the whole process is the average of the microscopic work $W^{\left[l\right]}.$

Figure 3c shows the fully quantum simulation realized on ibmqx2. With the five qubits, it is possible to realize at most two-step isothermal process, since the qubit resetting process is not permitted on ibmqx2. In the fully quantum simulation, the same circuit is implemented repetitively, and the excited state population $p_{e}\left(t_{j}\right)$ is obtained by measuring the state of the system qubit. The performed work for the simulated system is given by
(13)$$\bar{W}=\sum_{j=1}^{N}\left(\omega_{j}-\omega_{j-1}\right)p_{e}\left(t_{j-1}\right).$$

Since ibmqx2 does not allow the user to reset the state of the qubit, each elementary process requires new ancillary qubit(s). With the ability to reset the ancillary qubit, two (three) qubits are enough to complete the simulation with the hybrid simulation (fully quantum simulation) by resetting the ancillary qubit(s) at the end of each isochoric process. This control scheme is realized in Ref. [25] to simulate repetitive quantum channels on a single qubit.

## 4. Testing 1/τ Scaling of Extra Work

One possible application of the thermodynamic simulation is to test the 1/τ scaling of the extra work, where τ indicates the operation time of the finite-time isothermal process. In equilibrium thermodynamics, the performed work for a quasi-static isothermal process is equal to the change of the free energy ΔF [43]. The quasi-static isothermal process requires infinite time to ensure equilibrium at every moment. For a real isothermal process, irreversibility arises accompanied with the extra work. For a fixed control scheme, it is proved that the extra work decreases inverse proportional to the operation time at the long-time limit [44,45,46,47]. Such 1/τ scaling has been verified for the compression of dry air in the experiment [48].

The superconducting quantum circuit provides an experimental platform to study quantum thermodynamics. We demonstrate the scaling behavior of the extra work in finite-time isothermal process can be observed with the current experimental proposal. Here, the parameters of the simulated two-level system are chosen as $\gamma_{0}=1$ and β=1 for convenience. The energy spacing is tuned from $\omega_{0}=1$ to $\omega_{N}=2$ in *N* steps of elementary processes.

In Figure 4, the 1/N scaling of the extra work is shown with the ibmqx2 simulation results (Appendix A) for different operation time δτ=0.5 (blue dashed curve) and 10 (red solid curve). For large step number *N*, it is observed that the extra work is inverse proportional to the step number as $\bar{W}-\Delta F\propto 1/N$ [33,37,38]. The free energy difference of the final and the initial state, namely the performed work in the quasi-static isothermal process is
(14)$$\begin{array}{cc} \Delta F & =\omega_{N}-\omega_{0}-k_{B}T\ln\frac{1+e^{\beta \omega_{N}}}{1+e^{\beta \omega_{0}}}. \end{array}$$

With the chosen values of the parameters, the explicit value of the free energy difference is ΔF=0.186. Since the total operation time is τ=Nδτ, the 1/N scaling is consistent with the 1/τ scaling of the extra work in finite-time isothermal processes. The discrete isothermal processes are simulated on ibmqx2 for N=2,3 and 4 with the hybrid simulation (empty squares) and N=2 with the fully quantum simulation (pentagrams).

Figure 5 compares the simulation results on ibmqx2 and the numerical results. In (a) and (b), the work distribution of the hybrid simulation results (blue solid line) is compared to the exact numerical results (gray dashed line), with the operation time δτ=0.5 in (a) and δτ=10 in (b). For the hybrid simulation on ibmqx2, the maximum step number is N=4 with the five qubits. To mimic the random selection of the sub-channel, we simulate every possible selection of the sub-channels in the isochoric processes and measure the state populations of the system qubit. For each selection, the corresponding circuit is implemented on ibmqx2 for 8192 shots. The average work is obtained by summing the work in each selection with the corresponding probability $p_{\left\{K_{j}\right\}}=\prod_{j}p_{K_{j}}^{\left(j\right)}$ ($K_{j}=↑\;\mathrm{or}\;↓$). If the random selections of the sub-channels are possible, $p_{\left\{K_{j}\right\}}$ should be determined by the CRNG. Yet, here the probability of the selection $p_{\left\{K_{j}\right\}}$ is not implemented in the experiment but calculated with $p_{K_{j}}^{\left(j\right)}$ since the random selection of the two sub-channels cannot be implemented on ibmqx2.

Figure 5c,d show the excited state population of the system qubit for the fully quantum simulation of two-step isothermal process on ibmqx2. The operation time of each isochoric process is δτ=0.5 in (c) and δτ=10 in (d). The excited state populations $p_{e}\left(t_{j}\right)$ at $t_{j}=0,\;\delta \tau$ and 2δτ are obtained by implementing 40960 shots of the corresponding circuits. Compared to those of the numerical result (gray bar), the ibmqx2 simulation results (blue bar) are larger, since the noises in the quantum computer generally lead to a more mixed state. At the end t=2δτ of the process, the most quantum gates are used, and the absolute error reaches about 0.05. The fidelity between the simulation and the numerical results $F\left(t\right)=\sqrt{p_{g}^{\left(\mathrm{num}\right)}\left(t\right)p_{g}^{\left(\mathrm{sim}\right)}\left(t\right)}+\sqrt{p_{e}^{\left(\mathrm{num}\right)}\left(t\right)p_{e}^{\left(\mathrm{sim}\right)}\left(t\right)}$ is explicitly F(2δτ)=0.998 and 0.999 for the second step t=2δτ in (c) and (d), respectively.

The performed work in both the hybrid simulation and the fully quantum simulation is obtained according to Equations (Equation 12) and (Equation 13), as listed in Table 2. In Figure 4, the extra work in the ibmqx2 simulation results exceeds that of the numerical result due to the accumulated error in long circuits. The error mainly comes from the two-qubit gates, since the error probability in two-qubit gates (from $1.344\times 10^{-2}$ to $1.720\times 10^{-2}$) greatly exceeds that of single-qubit gates (from $3.246\times 10^{-4}$ to $2.164\times 10^{-3}$) [32]. The computing accuracy might be improved by using either quantum error correction or quantum mitigation [49]. Limited to the precision of operation on ibmqx2, the results deviate from the theoretical expectations.

The current simulation scheme have only considered the commutative Hamiltonian at different steps $[H\left(t_{j}\right),H\left(t_{j^{\prime }}\right)]=0$ and the adiabatic process as the quench with zero time $\delta \tau_{\mathrm{adi}}=0$. It can also be generalized to the discrete isothermal process with finite-time adiabatic processes, where the effect of the non-commutative Hamiltonian will increase the extra work [50]. For a generic adiabatic process, the unitary evolution of the two-level system should be simulated with the single-qubit gates on the system qubit. The off-diagonal elements of the initial density matrix cannot be neglected, since the changing ground and excited states lead to the interplay between the off-diagonal elements and the populations. Besides, the current simulation can be simplified for the ideal discrete isothermal process, where the perfect thermalization of the isochoric processes allows simulating each elementary process separately by preparing the equilibrium states at the beginning of the adiabatic processes [38].

With the limited number of qubits, we only show a few data points in Figure 4. It requires either more usable qubits or the ability of resetting to simulate the discrete isothermal process with a larger step number *N* in experiment. Another topic is to test the optimal control scheme [36]. For the given operation time τ, the control scheme is optimized to reach the minimum extra work. The lower bound of the extra work is related to the thermodynamic length [44,46,51,52], which endows a Riemann metric on the control parameter space. The current experimental proposal might also be utilized to measure the thermodynamic length of the isothermal process for the two-level system.

## 5. Conclusions

We show an experimental proposal to simulate the finite-time isothermal process of the two-level system with the superconducting quantum circuits. Two methods, the hybrid simulation, and the fully quantum simulation, are proposed to realize the generalized amplitude damping channel. Assisted by the classical random number generator or the quantum superposition, the hybrid or the fully quantum simulation can simulate an *N*-step isothermal process with N+1 or 2N+1 qubits, respectively.

We have used the quantum computer of IBM (ibmqx2) to demonstrate the simulation of the discrete isothermal processes, which have been realized for four steps with the hybrid simulation and two steps with the fully quantum simulation. If more steps of elementary processes can be realized experimentally, the 1/τ scaling of the extra work can be tested by the thermodynamic simulation on the universal quantum computer.

## Figures and Tables

**Figure 1 entropy-23-00353-f001:**
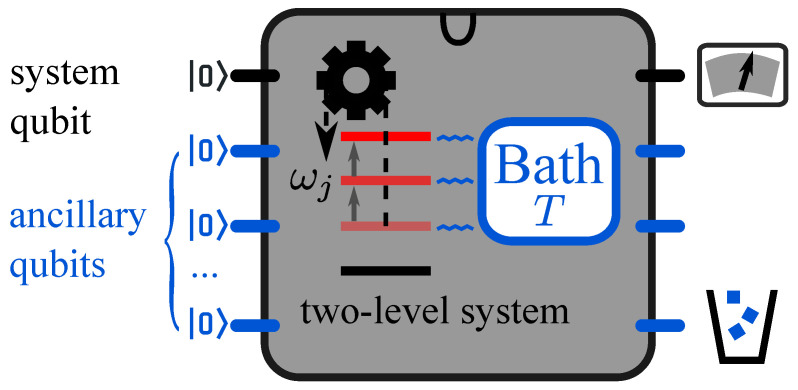
Simulation of the isothermal process on the superconducting quantum computer. The finite-time isothermal process is divided into series of piecewise adiabatic and isochoric processes. In the adiabatic process, the energy of the two-level system is tuned with the switched-off interaction between the system and the thermal bath. In the isochoric process, the interaction is switched on with the unchanged energy spacing $\omega_{j}$. One qubit represents the simulated two-level system, and the ancillary qubits play the role of the thermal bath at the temperature *T*. After implementing the quantum circuit, the system qubit is measured to obtain the internal energy.

**Figure 2 entropy-23-00353-f002:**
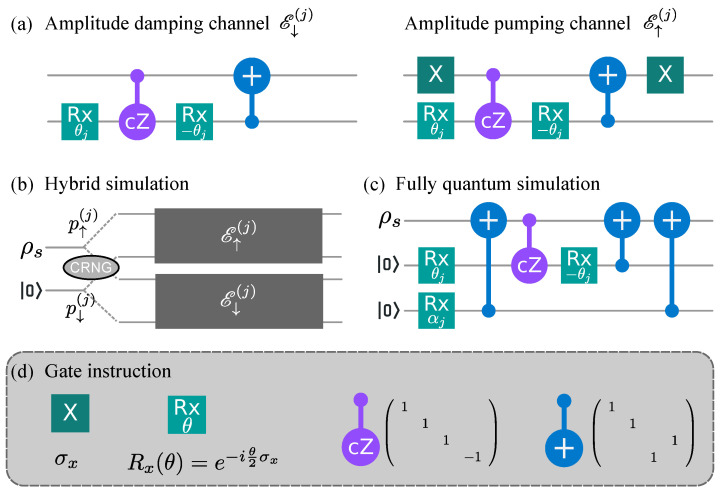
The quantum circuits in one elementary process. (**a**) The amplitude damping (pumping) channel $E_{↓}^{\left(j\right)}$ ($E_{↑}^{\left(j\right)}$) in the hybrid simulation. (**b**) One elementary process in the hybrid simulation. The selection of the two sub-channels is realized by the classical random number generator. (**c**) One elementary process in the fully quantum simulation. The selection of the two sub-channels is assisted by another ancillary qubit. (**d**) Instruction of gates in the current simulation.

**Figure 3 entropy-23-00353-f003:**
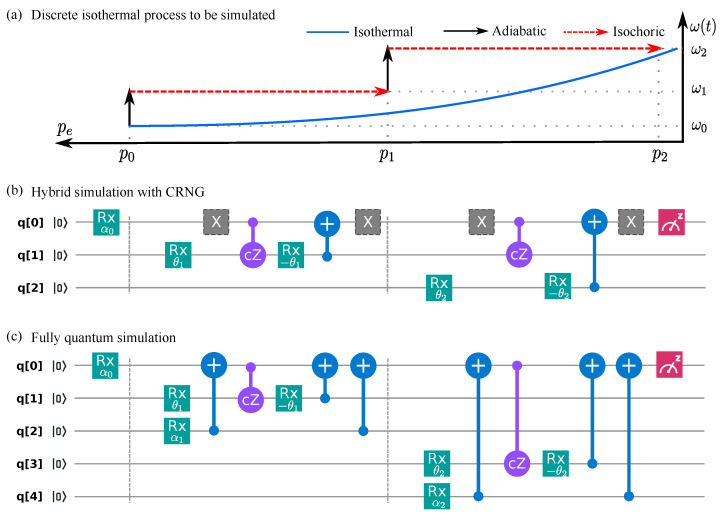
The circuit of the two-step isothermal process on ibmqx2. (**a**) Excited state population-energy ($p_{e}-E$) diagram. (**b**) The circuit for the hybrid simulation. In each elementary process, the X gate is (or not) implemented for the sub-channel selected as the amplitude pumping (damping) channel according to the classical random number. Each elementary process requires another ancillary qubit. (**c**) The circuit for the fully quantum simulation. Each elementary requires two ancillary qubits.

**Figure 4 entropy-23-00353-f004:**
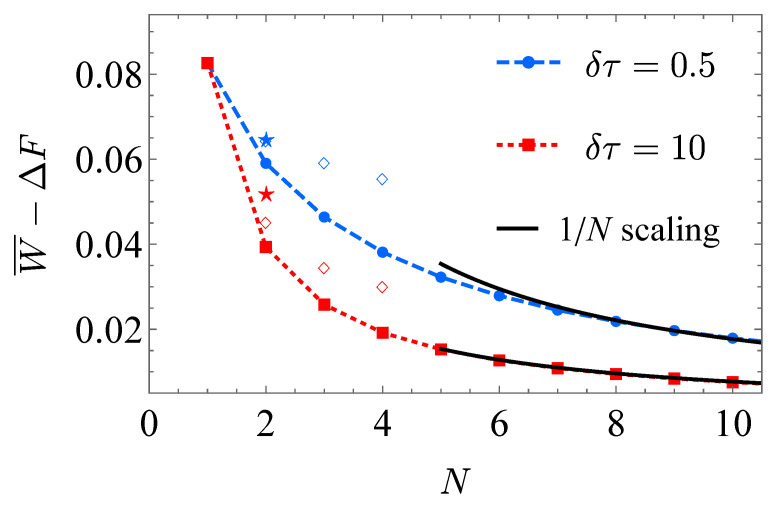
1/N scaling of the extra work for the discrete isothermal process. The operation time of each isochoric process is set as δτ=0.5 (blue dashed curve) or 10 (red solid curve). The ibmqx2 simulation results for N=2,3 and 4 are plotted. The empty squares present the results by the hybrid simulations, and the pentagrams for the fully quantum simulation. The 1/N scaling is shown by the solid black curve.

**Figure 5 entropy-23-00353-f005:**
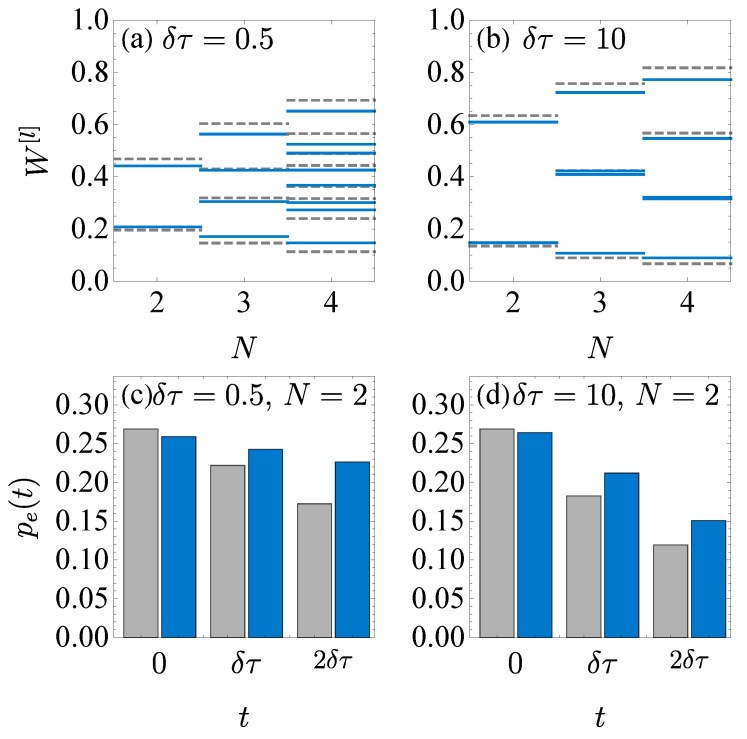
Comparison of the ibmqx2 simulation and the numerical results. (**a**,**b**) show the microscopic work in the hybrid simulation with the step number N=2,3 and 4. The ibmqx2 simulation result (blue solid line) is compared with the numerical result (gray dashed line). (**c**,**d**) show the excited state population $p_{e}\left(t\right)$ at each step in the fully quantum simulation of the two-step isothermal process. The ibmqx2 simulation results (blue bar) are compared to the numerical results (gray bar).

**Table 1 entropy-23-00353-t001:** The discrete isothermal process to be simulated and the two simulation methods, the hybrid simulation and the fully quantum simulation

	To be Simulated:	Simulation	
	Discrete Isothermal Process	Hybrid Simulation with CRNG	Fully Quantum Simulation
Adiabatic process	U[R(t)], $t\in [t_{j-1},t_{j}]$	The unitary evolution is realized with the virtual tuning on the system Hamiltonian.	
Isochoric process	System relaxation in Equation (Equation 2)	Generalized amplitude damping channel $E_{\mathrm{GAD}}^{\left(j\right)}$ with the classical random number generation	Generalized amplitude damping channel $E_{\mathrm{GAD}}^{\left(j\right)}$ with an additional qubit at the state $\cos\left(\alpha_{j}/2\right)|0\rangle +i\sin\left(\alpha_{j}/2\right)|1\rangle$
Parameters	Duration: $\delta \tau =t_{j}-t_{j-1}$ Temperature: *T*	$\cos\theta_{j}=\exp\left[-\frac{\gamma_{0}\delta \tau }{2}\coth\left(\frac{\beta \omega_{j}}{2}\right)\right]$	$\cos\theta_{j}=\exp\left[-\frac{\gamma_{0}\delta \tau }{2}\coth\left(\frac{\beta \omega_{j}}{2}\right)\right]$ $\cos\left(\alpha_{j}/2\right)={\left[p_{↓}^{\left(j\right)}\right]}^{1/2}$

**Table 2 entropy-23-00353-t002:** The performed work obtained by the ibmqx2 simulation and the numerical results.

	*N*	δτ=0.5		δτ=10	
		${\bar{W}}_{\mathrm{ibmqx}2}$	${\bar{W}}_{\mathrm{exact}}$	${\bar{W}}_{\mathrm{ibmqx}2}$	${\bar{W}}_{\mathrm{exact}}$
Hybrid simulation	2	0.251	0.245	0.232	0.226
	3	0.246	0.233	0.221	0.212
	4	0.243	0.224	0.217	0.206
Fully quantum simulation	2	0.251	0.245	0.238	0.226

## Data Availability

Not applicable.

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
