# Peer review of "Simulating Finite-Time Isothermal Processes with Superconducting Quantum Circuits"

_entropy, 2021, doi:10.3390/e23030353_

Round 1

Reviewer 1 Report

This manuscript considers the simulation of finite-time isothermal processes in a superconducting quantum circuit. The idea is rather interesting, given the increasing interest in these processes in the community. While the discussed protocols are essentially classical (the driven Hamiltonian commutes with itself at all times), some ideas presented here could be adapted to more challenging quantum processes. 

Having said that, I also think the article needs to be substantially improved to merit publication in Entropy. Some comments follow:

  1. In general, the writing is not very clear, and the English needs to be improved. In particular, the abstract can be substantially improved.
  2.  In my opinion, one of the main limitations of this article is that one can only simulate very low N<5. Given that the simulated process is essentially classical, it really seems that the circuit/implementation could be improved to reach higher N. In particular, the authors could consider a similar protocol as the one considered in Phys. Rev. Research 2, 023377 (2020) to the present situation where the thermalisation is not complete at each step. In this case, one could simulate each step independently of the other ones (so that only 2 or 3 qubits are used for each step), and then combine the obtained statistics for each step. In particular, the outcome of the energy measurement in the "jth" step is used as the intial state of the "(j+1)th" step. Note that, since we are dealing with a classical process, doing energy measurements after every step  will not change the statistics of the next measurements. For the simulation, one simply needs to be careful to prepare the state of the "jth" step given the result of the previous measurement ("j-1th" step). And then repeat the process many times to obtain good statistics. This would provide a systematic way to reach high N, and hence check the 1/N scaling. 
  3. There is a noticeable discrepancy between the simulation and the theoretical expectation (Fig. 4). The authors mention in line 289 that this is due to noise in quantum computer. Can this be simulated in the theoretical model to reach a better agreement?
  4. Regarding literature, there are several relevant references that could be added. In particular, these kind of discrete processes to model isothermal processes have been considered in the past: for the first time in: Nulton et al, The Journal of Chemical
    Physics, 83:334, (1985), and more recently in New Journal of Physics 15 (3), 033022 (2013), Quantum 3, 153 (2019), to name a few. See also   Phys. Rev. Research 2, 023377 (2020) and references therein. Also, regarding the 1/\tau scaling, this has been also proved for open quantum systems in Phys. Rev. Lett. 119, 050601 (2017). 

In summary, this paper presents a rather interesting idea but the results (obtained for very small N<5) are not conclusive. Given that the simulated process is, in essence, classical, it appears that a modification of the protocol would enable reaching higher N with similar resources. In view of this fact, and that overall the article is not very clear in its presentation, I recommend reconsideration for Entropy after major revision. 

Reviewer 2 Report

In this paper the authors simulate finite-time isothermal processes of a two-level system using a universal quantum computer (the ibmqx2) adopting a discrete-step method, whereby the isothermal process is divided into series of elementary processes consisting of an adiabatic process and an isochoric process. The dynamics is realised using quantum gates without directly tuning physical parameters (a constraint imposed by ibmqx2). The 1/\tau scaling of the extra work has also been addressed in the simulations.

The subject of the paper (simulations using a universal quantum computer) is very interesting and timely, the procedures used for the simulations are clearly described. The paper is sound and well written, the results are original and interesting in the field of quantum information in condensed matter. In conclusion, I believe that the paper deserves publication in Entropy.

Author Response

———————————

Summary of the main changes

———————————

1. We have revised the language of the manuscript.

2. The references suggested by the other referee are added in the revised manuscript.

———————————

Reply

———————————

We would like to thank the referee for carefully reading and giving the conclusion about this manuscript.

Round 2

Reviewer 1 Report

The present version has improved, especially in terms of presentation, writing, and clarity. 

I understand that unfortunately some of the scientific issues that I raised cannot be addressed due to the inability to access the quantum simulator.

Still, as I mention in my previous report, this is a rather interesting idea, which might motivate better simulations (with larger N), and more general protocols, in the future. In view of this fact, I recommend this paper for publication in Entropy. 

Author Response

We would like to thank the referee for reading and recommending the publication of the current manuscript.